# Molecule-based nonlinear optical switch with highly tunable on-off temperature using a dual solid solution approach

Shi-Yong Zhang[1,2,4], Xia Shu[2,4], Ying Zeng[3,4], Qing-Yan Liu [1], Zi-Yi Du [1✉], Chun-Ting He[1✉], Wei-Xiong Zhang [3✉] & Xiao-Ming Chen[3]

Nonlinear optical switches that reversibly convert between on/off states by thermal stimuli are promising for applications in the fields of photoelectronics and photonics. Currently one main drawback for practical application lies in the control of their switch temperature, especially for the temperature range near room temperature. By mixed melting treatment, here we describe an alloy-like nonlinear optical switch with tunable switch temperature via a dual solid solution approach within the coordination polymer system. We initially prepare a coordination polymer ($i$-PrNHMe$_2$)[Cd(SCN)$_3$], which functions as a high-contrast thermo-responsive nonlinear optical switch originating from a phase transition at around 328 K. Furthermore, by taking advantage of a synergistic dual solid solution effect, the melt mixing of it with its analogue (MeNHEt$_2$)[Cd(SCN)$_3$], which features an unequal anionic chain templated by an isomeric ammonium, can afford coordination polymer solid solutions with switch temperatures that are tunable in a range of 273–328 K merely by varying the component ratio.

[1] College of Chemistry and Chemical Engineering, MOE Key Laboratory of Functional Small Organic Molecule, Jiangxi Normal University, Nanchang 330022, China. [2] College of Chemistry and Chemical Engineering, Gannan Normal University, Ganzhou 341000, China. [3] MOE Key Laboratory of Bioinorganic and Synthetic Chemistry, School of Chemistry, Sun Yat-Sen University, Guangzhou 510275, China. [4] These authors contributed equally: Shi-Yong Zhang, Xia Shu, Ying Zeng. ✉email: ziyidu@gmail.com; hct@jxnu.edu.cn; zhangwx6@mail.sysu.edu.cn

Nonlinear optical (NLO) materials are of great importance due to their pivotal role in the fields of photoelectronics and photonics. In recent years, the research on NLO switches, of which quadratic NLO effect such as second-harmonic generation (SHG) can be on-off switched under external physical stimuli, have aroused intense interest[1–9]. So far, one most feasible method is to tweak the conformation or orientation of NLO-chromophore molecules in the crystal structures by thermal stimulus, with a mechanism related to structural phase transition, which can realign the NLO-active moieties to alter the SHG signal, especially those with the transition from a non-centrosymmetric space group to a centrosymmetric one[2,10–20]. Following this strategy, the thermoresponsive NLO switches within the molecule-based systems, typically represented by the coordination polymers (CPs), can be achieved with many advantages such as the designable and flexible structures, the relatively mild preparation condition as well as appropriate on/off contrast of SHG signal. Nevertheless, most of the reported thermoresponsive NLO switches do not have the tunability on the regulation of their on-off temperatures ($T_S$), especially for the temperature range above and below the room temperature, which is a key issue for practical application.

To resolve the above issue, the formation of CP solid solutions or alloys that retain a similar structure to the NLO-switchable parent phase may be an efficient and convenient method, of which the $T_S$ could be finely tuned by the varied ratio of such binary system. According to the similarity-intermiscibility theory, one key requirement for the syntheses of CP solid solution is that the two mixed CPs should be structurally similar, which avoids a notable increase of the crystal cohesive energy. So, from the view of thermodynamics, entropy increase usually is a driving force for the formation of CP solid solutions. On the other hand, as is widely used, one most effective procedure for the preparation of solid solution is via the mixed melting-cooling treatment. However, by comparison with metal alloys and more conventional materials, there have been few reports on the preparations of CP-based solid solutions via mixed melting, mainly owing to the near-ubiquitous irreversible thermal decomposition of CPs upon heating to moderate temperatures[21–23]. To find a CP structure that can be melted before decomposition, we noticed that low-dimensional negatively charged CPs templated with charge-balanced guest cations may be good candidates[24–26]. Meanwhile, the polar guest cations may also function as second-order NLO chromophores.

Up to now, the reported CP solid solutions are still scarce, usually realized through the one-pot reaction in solution or mechanical grinding, using a simple one-site doping or single-component substitution strategy, such as mixed metal ions, mixed ligands, or mixed ammonium, to tune the related properties[27–37]. These solid solutions often have properties that are between those of the pure components. For instance, the phase transition temperatures of the phase-changeable CP solid solutions are often between those of the unmixed components (*Modes 1* and *2* shown in Fig. 1). To expand the tunable scope inspired by the mixed-cation lead mixed-halide perovskite[38], we conceive that a synchronous dual mixing strategy, i.e., the *Mode 3*, may shift the tunable scope to lower temperature range, which is somewhat similar to the melting point reduction effect of common metal alloys due to the weakening interatomic forces.

Bearing the aforementioned considerations in mind and adopting the strategy of *Mode 3* in Fig. 1, we thus put forward a trial project shown in Fig. 2, and our preliminary research endeavors yielded two such CPs, ($i$-PrNHMe$_2$)[Cd(SCN)$_3$] (**1**) and (MeNEt$_2$)[Cd(SCN)$_3$] (**2**), both of which have same empirical formula and undergo reversible solid–solid and solid–liquid phase transitions above room temperature. Importantly, **1** can not only function as a high-contrast thermoresponsive NLO switch owing to the solid–solid phase transition, but also form CP solid solutions with **2** which exhibit a highly adjustable on-off temperature of the NLO effect merely by varying the component ratio.

## Results

**Thermal analyses for CPs 1 and 2**. The thermogravimetric curves of **1** and **2** are very similar, and indicate that both of them are stable up to about 416 K, whereupon a decomposition process occurs (Supplementary Fig. 1). Differential scanning calorimetry (DSC) curves of **1** and **2** both show two pairs of endothermic/exothermic peaks at heating/cooling runs (Fig. 3), revealing that each of them undergoes two reversible phase transitions above room temperature. Variable-temperature powder X-ray diffraction (PXRD) analyses as well as the melting point test further confirm that the first phase transition is a solid–solid one (Supplementary Fig. 2), whereas the second is a solid–liquid one. The DSC curves indicate that **2** has a little higher $T_{C(\text{solid–solid})}$ and a slightly lower $T_{C(\text{solid–liquid})}$ in comparison with those of **1**. For convenience, we label the solid phases below and above the $T_{C(\text{solid–solid})}$ as **α** and **β** phase, respectively. For such solid–solid phase transitions, the sharp peak shapes, prominent enthalpy changes (27.7/–26.6 J g$^{-1}$ for **1** and 18.9/–18.7 J g$^{-1}$ for **2** during the heating/cooling process), and somewhat wide thermal hysteresis between the heating and cooling runs (~6.2 K for **1** and ~12.7 K for **2**) reveal the typical characteristics of a first-order phase transition. On the other hand, the enthalpy changes of the solid–liquid phase transitions (60.8/–59.3 J g$^{-1}$ for **1** and 63.7/–61.8 J g$^{-1}$ for **2** during the heating/cooling process) are about 2.2 and 3.4 times, respectively, in comparison with their corresponding solid–solid ones.

**The thermoresponsive NLO switching of CP 1**. The preliminary NLO effect tests were first performed for **1** and **2** at room temperature using the Kurtz and Perry method[39], showing that **1** is SHG-active whereas **2** is SHG-inactive. Further experiment for **1** proved that such effect is phase-matchable (Supplementary

| | I | + | II | Solid Solution → | III | $T_{\text{phase transition}}$ |
|---|---|---|---|---|---|---|
| *Mode 1* | AB | | A′B | | (AA′)B | $T_I < T_{III} < T_{II}$ |
| *Mode 2* | AB | | AB′ | | A(BB′) | |
| *Mode 3* | AB | | A′B′ | | (AA′)(BB′) | $T_{III} < T_I, T_{II}$ |

**Fig. 1 Three mixing modes for the formation of phase-changeable CP solid solutions.** A single-component substitution method (*Modes 1* and *2*) and a synchronous dual mixing strategy (*Mode 3*).

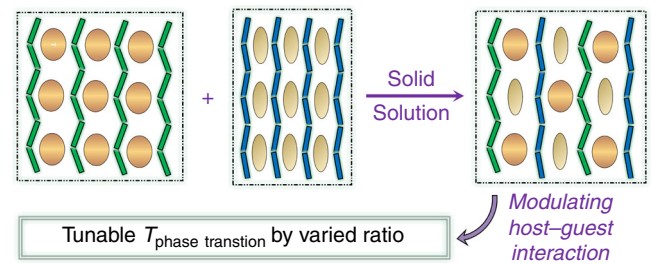

Tunable $T_{\text{phase transtion}}$ by varied ratio

Modulating host–guest interaction

**Fig. 2 Molecular-level mixing of two phase-changeable 1D CPs using a dual mixing strategy.** Unequal anionic chains are represented by zigzag chains with different bending degree, and distinct guest ammonium templates are represented by ellipsoids with different size.

Fig. 3). A comparison of the second-harmonic signal produced by samples of **1** and commercialized KDP ($KH_2PO_4$) in the same particle range of 150~210 mm revealed that it exhibits a SHG efficiency ~1.5 times that of KDP. In consideration of its structural phase transition, in situ variable-temperature NLO effects were studied in the range of 305~345 K. As shown in Fig. 4, upon heating, its SHG signals were maintained in **α** phase, while higher temperature over $T_C$ leads to a complete obliteration of SHG effects. In its NLO-inactive state at **β** phase, no SHG signal was detected, except for quite weak noise errors (inset in Fig. 4). The subsequent cooling process confirmed that such switch is reversible, corresponding to the DSC measurements. The NLO switching contrast of **1**, defined as the ratio of SHG intensities at high-NLO and low-NLO states, was found to be ~74. Such a value is higher than those for most solid-state materials reported to date[13,20], demonstrating that **1** is a good candidate as NLO-switching material.

**Structural analyses of CP 1.** As is known, an essential requirement for crystalline SHG-active materials is that they must be assembled into a noncentrosymmetric structure. To understand the NLO-switching mechanism of **1**, firstly single-crystal X-ray diffraction was performed for **1** at two different temperatures, revealing that it crystallizes in the polar space group $Cmc2_1$ at **α** phase and in the centrosymmetric space group $P6_3/mmc$ at **β** phase (Supplementary Table 1). The crystal

structures can be roughly described as a hexagonal perovskite-like structure, which is similar to those of the inorganic $BaNiO_3$ and the organic–inorganic (Pyrrolidinium)$MnBr_3$[40], with a general formula of $ABY_3$ (the valence ratio of A:B:Y is +1: +2:−1). The asymmetric unit of **1α** contains a half-occupied $Cd^{2+}$ cation, one full and a half-occupied $SCN^-$ anions as well as a half-occupied ($i$-PrNHMe$_2$)$^+$ cation (Supplementary Fig. 4). The half-occupied Cd(1) ion, locating in a crystallographic mirror plane, is octahedrally coordinated to three N and three S atoms from six $SCN^-$ anions, in a *fac*-configuration. The two types of $SCN^-$ anions in **1α**, with one locating in a mirror plane and the second occupying a general position, both act as end-to-end bridging ligands between two $Cd^{2+}$ ions. The bridging of the $Cd^{2+}$ ions through these $SCN^-$ anions leads to an infinite $\{[Cd(SCN)_3]^-\}_\infty$ chain, in which each pair of neighboring $Cd^{2+}$ ions is triply bridged by three $SCN^-$ anions, and the coordination atoms between two neighboring $Cd^{2+}$ ions are "$1S + 2N$" on one side and "$1N + 2S$" on the other side (Fig. 5a). The overall crystal structure of **1α** can be viewed as an approximate hexagonal packing of the parallelly aligned $\{[Cd(SCN)_3]^-\}_\infty$ chains that run along the *c*-axis, extending along the *ab*-plane, with charge-balanced discrete ($i$-PrNHMe$_2$)$^+$ cations filling in the inter-chain space (Fig. 5a and Supplementary Fig. 5). It is noted that the ($i$-PrNHMe$_2$)$^+$ cation in **1α** is two-fold disordered, related by a crystallographically imposed mirror symmetry.

From **1α** to **1β**, one significant structural change of the crystal is the highly disorder of all components, suggesting that such phase transition belongs to an order-disorder type. As shown in Fig. 5b and Supplementary Fig. 6, the host $\{[Cd(SCN)_3]^-\}_\infty$ chain and guest ($i$-PrNHMe$_2$)$^+$ cation turn six- and twelvefold disordered, respectively, both related by a crystallographically imposed symmetry. What's more, the synergetic disorder of the host and guest components results in the transform from a noncentrosymmetric space group to a centrosymmetric one, which coincides well with the observed NLO-switching phenomenon. Considering the highly disordered state of all components, the **1β** phase can be regarded as a transitional "melt-like" state. From the viewpoint of symmetry breaking, a total symmetry decrease from twenty four symmetric elements ($E$, $2C_6$, $2C_3$, $C_2$, $3C_2'$, $3C_2''$, $i$, $2S_3$, $2S_6$, $\sigma_h$, $3\sigma_v$, $3\sigma_d$) to four ($E$, $C_2$, $2\sigma_v$) during the transition **β→α** (Supplementary Fig. 7), denoting that this transition is a ferroelastic one with an Aizu notion of $6mmmFmm2$[41].

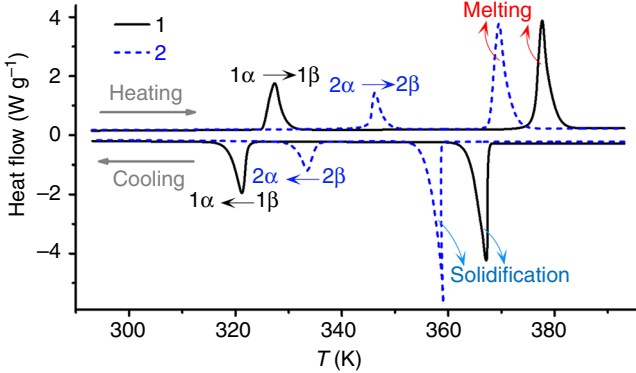

**Fig. 3 Differential scanning calorimetry of coordination polymers.** DSC measurements for CPs **1** and **2** recorded on a heating−cooling cycle.

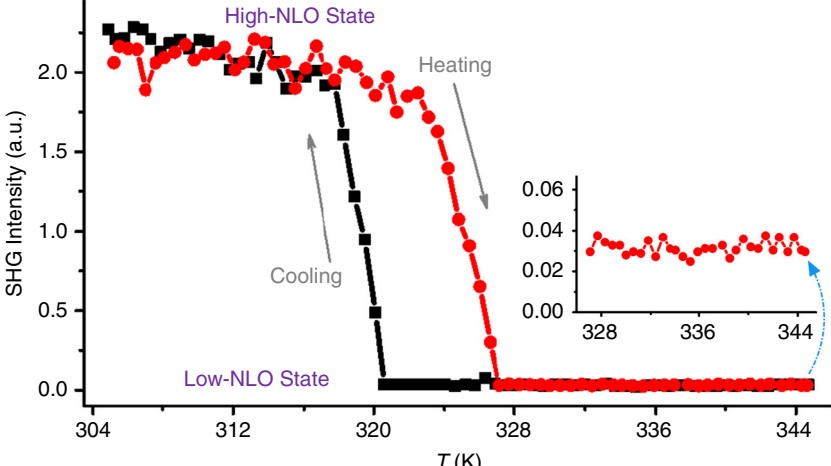

**Fig. 4 Nonlinear optical switching in coordination polymers.** Switchable conversion of SHG signals between high-NLO and low-NLO states of CP **1**.

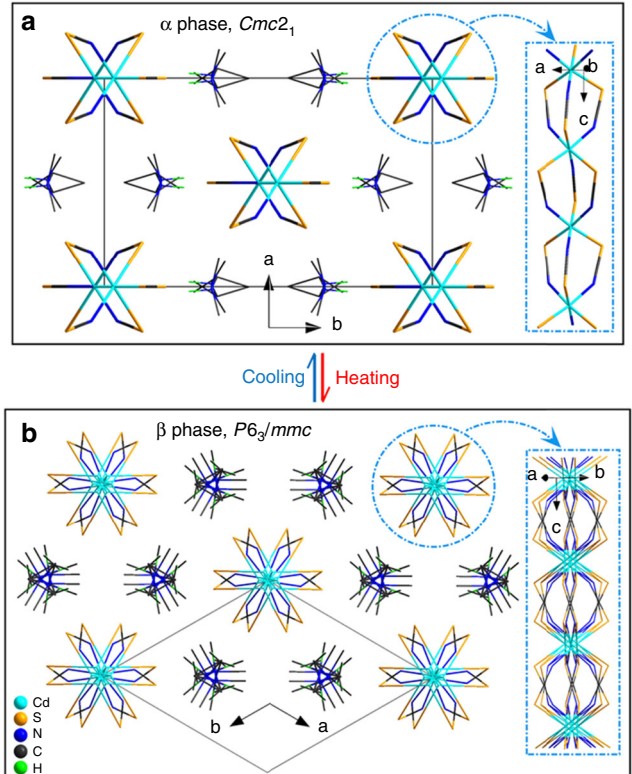

**Fig. 5 Comparison of the crystal structures between 1α and 1β phase. a, b** View of the packing structures of **1α** at 296 K and **1β** at 348 K. The disordered bonds are represented by thin lines. The C-bound H atoms have been omitted for clarity.

**Molecular dynamics simulations of CP 1**. To further gain a direct insight into the microscopic dynamic characteristics of **1** and uncover the macroscopic NLO-switching mechanism, we performed molecular dynamics (MD) simulations at 296 and 363 K, respectively, based on the corresponding unit-cell parameters of **α/β** phase (keeping the volumes unchanged). As shown in Supplementary Fig. 8a–i, the orientation of the ammonium N–H bond in **1α** phase, which can be roughly viewed as the polarizability tensor of a second-order NLO chromophore, i.e., ($i$-PrNHMe$_2$)$^+$ ion, is almost invariable over the simulation time, and no rotation is observed for the host {[Cd(SCN)$_3$]$^-$}$_\infty$ chain (Supplementary Fig. 8j). The two-fold disorder of the ($i$-PrNHMe$_2$)$^+$ ion in **1α** phase is static in essence. In contrast, the MD simulation result shows that the N–H bond in **1β** phase constantly and markedly changes its orientation during the simulation process (Fig. 6a–i). Thus, the statistical superposition effect of the microscopic polarizabilities for these dynamically disordered NLO chromophores within the crystal leads to a vanishing of the macroscopic hyperpolarizability in **β** phase. Meanwhile, a certain degree of dynamic rotation can be found for the host {[Cd(SCN)$_3$]$^-$}$_\infty$ chain (Fig. 6j). Overall, the MD simulation results coincide well with the above-mentioned single-crystal structure analyses, and perfectly explain how the polar space group in **α** phase can transform to a centrosymmetric one.

**Structural analyses of CP 2**. For **2**, it can be viewed as an isomeric analogue of **1**. Different to the polar packing of **1α**, **2α** crystallizes in a centrosymmetric space group $P2_1/c$, with the asymmetric unit containing two types of Cd$^{2+}$ ions, six unique SCN$^-$ anions as well as two crystallographically independent (MeNHEt$_2$)$^+$ cations (Supplementary Fig. 9). The bridging of adjacent Cd$^{2+}$ ions through the end-to-end SCN$^-$ anions also

leads to an infinite {[Cd(SCN)$_3$]$^-$}$_\infty$ chain, which is topologically similar to that of **1α** (Supplementary Fig. 10). Although the Cd$^{2+}$ ion in **2α** is also 3N3S hexa-coordinated as that in **1α**, one subtle difference is that the three coordinated N/S atoms are arranged in a *mer*-configuration rather than a *fac*-configuration, resulting in a smaller bending degree of the zigzag anionic chain (Supplementary Fig. 11). This difference suggests that the shape and size of the guest ammonium ion have a strong template effect on the formation of these host anionic chains. It is noted that if N,N'-dimethylimidazolium is employed as the guest cation, a distinct anionic chain with 4N2S and 2N4S coordination environment can be formed[42]. From the crystallography perspective, the {[Cd(SCN)$_3$]$^-$}$_\infty$ chain and each pair of disordered ($i$-PrNHMe$_2$)$^+$ cations in **1α** both are related by a crystallographically imposed mirror symmetry, while the isomeric {[Cd(SCN)$_3$]$^-$}$_\infty$ chain and (MeNHEt$_2$)$^+$ cation in **2α** both locate at general position. Because of the seriously disordered state of **2β** phase, the single-crystal X-ray data collection at this phase cannot be performed. However, the experimental PXRD pattern at 358 K shows that this phase still retains a crystalline state, and the Pawley refinement (Supplementary Fig. 12) reveals the presence of a monoclinic unit cell: $a = 15.729(1)$ Å, $b = 9.0241(1)$ Å, $c = 10.7840(8)$ Å, $V = 1508.79(1)$ Å$^3$ (residuals $R_p = 3.17\%$, $R_{wp} = 4.35\%$), with $C2/c$ as the most possible space group. The molecular volume unit ($V_{cell}/Z$) expands ~5.6% as the temperature increases from 296 to 358 K.

**X-ray absorption near-edge structure analyses**. Upon further heating of **1β** or **2β**, a melting process can be observed (for instance, see Supplementary Movie 1). To monitor the detailed changes of the coordination bonds during the heating process, an in situ variable-temperature, synchrotron-based spectroscopy of X-ray absorption near-edge structure (XANES) was conducted. Supplementary Fig. 13 shows the XANES at the Cd $K$-edge for **1** and **2** at different temperatures, with Cd foil, [Cd(H$_2$O)$_6$]$^{2+}$ in aqueous solution (Cd$_{aq}$) and CdS as the reference samples (the valence state of Cd foil is zero). As guided by the arrow, when the Cd $K$-edge shifts to right, the valence state of Cd becomes higher. Thus, the order of oxidation state is as following: Cd-foil < **1** ≈ **2** ≈ CdS < Cd$_{aq}$. The extended X-ray absorption fine structure (EXAFS) fitting at the Cd $K$-edge (Supplementary Fig. 14) afforded more useful information such as the local coordination numbers and the bond distances about the Cd atom (Supplementary Table 4). It is clearly demonstrated that the coordination bonds around the Cd atoms show no breaking in the melting state, keeping six-coordinated. However, a notable change is that the Cd–S bonds shorten from ~2.72 Å (at 296 K) to ~2.59 Å (at 398 K) for **1**, and similar change is also observed for **2**. To achieve a high resolution in both $k$ and $R$ spaces, the wavelet transform was employed for further EXAFS analysis[43]. In comparison with the wavelet transform contour plots of the referenced Cd$_{aq}$ and CdS samples, those of **1** and **2** (Fig. 7, Supplementary Fig. 15) not only unambiguously distinguishes the coordination of both S/N atoms, but also confirms the shortening of the Cd–S bond in the melting state. This abnormal shortening of bond length at higher temperature can be ascribed to the tension contraction along the chain direction. In the solid state, the interaction between the guest ammonium and the host chain makes the chain tighten. When turning to melting state, the weakening host-guest interaction loosens the chain tension to some extent.

**Solid solutionizing and tunable NLO switch temperature**. By and large, the packing structures of **1** and **2** can be described as two types of parallelly arranged {[Cd(SCN)$_3$]$^-$}$_\infty$ chains, each with charge-balanced ($i$-PrNHMe$_2$)$^+$ or (MeNHEt$_2$)$^+$ cations filling in their inter-chain region, respectively. In addition, a

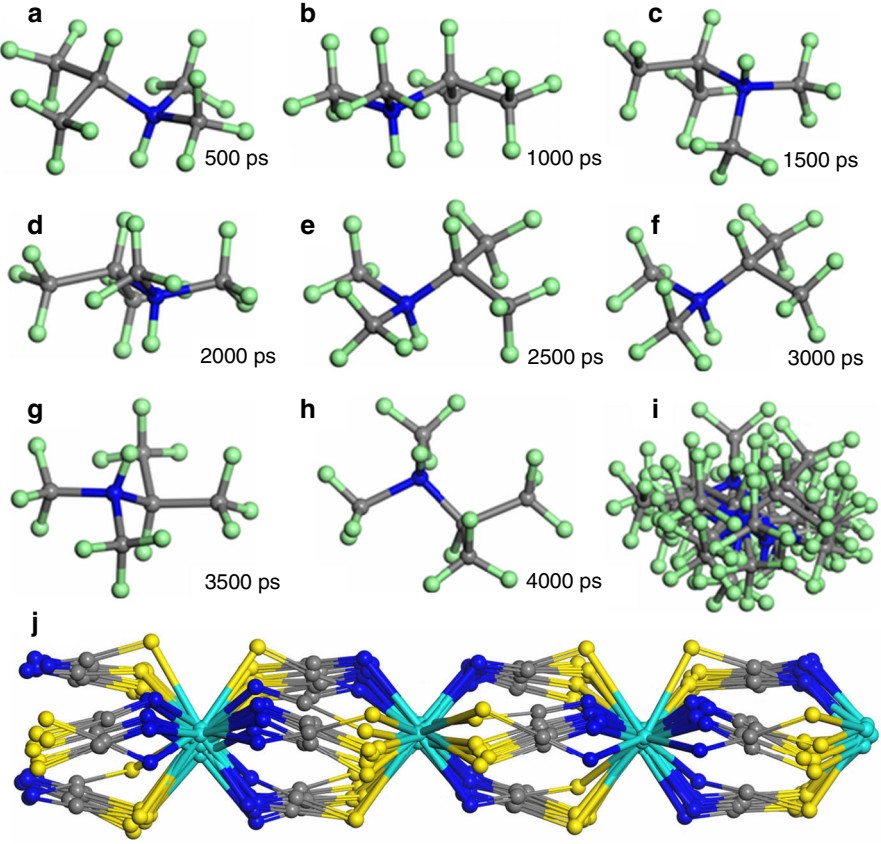

**Fig. 6 Constant-volume and temperature dynamic simulation for CP 1 at 363 K. a–h** Local snapshots of a ($i$-PrNHMe$_2$)$^+$ cation over the simulation time, showing the orientation changes. **i** Overlapping maps of the snapshots of (**a**)–(**h**). **j** The corresponding local snapshots of a {[Cd(SCN)$_3$]$^-$}$_\infty$ chain are also overlapped, to display its dynamics.

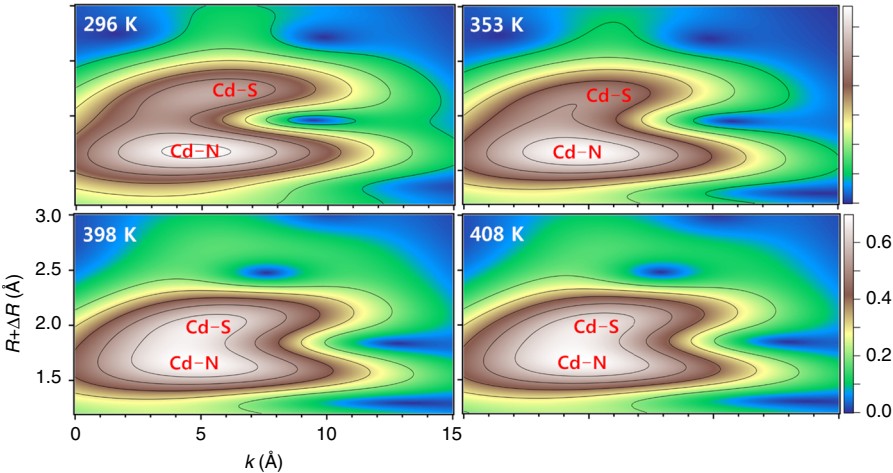

**Fig. 7 Wavelet transform EXAFS of CP 1 at different temperatures.** The heights of the peaks are represented by different colors.

sticky melting state of them can be observed upon further heating of the **1β** or **2β** phase. These structural characteristics are very beneficial for the preparation of CP solid solutions through the "mixing → melting → cooling" procedure. Hence, by utilizing the structural similarities as suggested in Fig. 2, we successfully prepared the continuous CP solid solutions of **1** and **2**, which can be modulated in an arbitrary mixing ratio (For experimental details, see the Supporting Information). The experimental PXRD patterns of these CP solid solutions confirm that they are not the mechanical mixing of **1** and **2** but all form a single phase (Fig. 8). Furthermore, the Pawley refinements on these observed PXRD data (Supplementary Fig. 16) reveal that the room temperature unit-cell parameters of {**1**$_{0.99}$**2**$_{0.01}$}, {**1**$_{0.976}$**2**$_{0.024}$} and {**1**$_{0.952}$**2**$_{0.048}$} are similar to those of **1α** phase, while {**1**$_{0.667}$**2**$_{0.333}$} and {**1**$_{0.50}$**2**$_{0.50}$} are similar to those of **1β** phase (Supplementary Table 5). As for {**1**$_{0.875}$**2**$_{0.125}$}, the room temperature PXRD

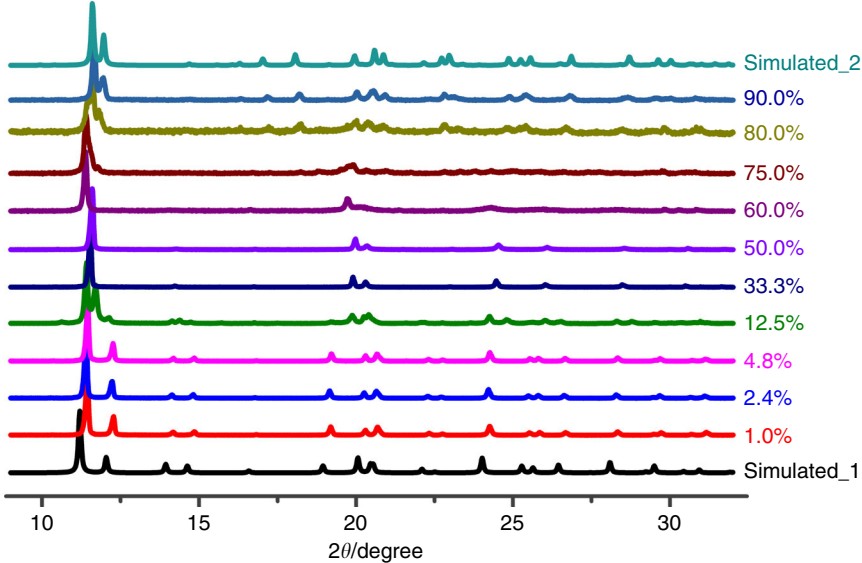

**Fig. 8 Room temperature PXRD patterns for the CP solid solutions {$1_{1-x}2_x$}.** The mixing ratio ($x$) of CP **2** in these solid solutions increases from 1.0 to 90.0%.

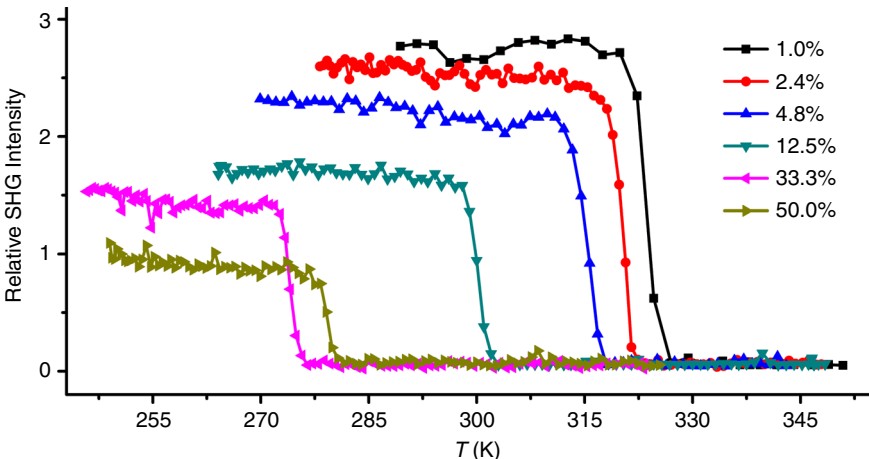

**Fig. 9 Variable-temperature SHG intensity of CP solid solutions {$1_{1-x}2_x$}.** The measurements were performed on their polycrystalline samples upon a heating process. The mixing ratio ($x$) of CP **2** ranges from 1.0 to 50.0%.

pattern is recorded just during the phase transition, which can be further identified by the DSC measurement (vide infra).

The impact of solid solution on the phase transitions was first analyzed using DSC measurement (Supplementary Figs. 17 and 18). With increasing proportion of $x$, the $T_{C(solid–solid)}$ decreases approximately linearly from ~328 K for pure **1** to ~273 K for solid solution {$1_{0.667}2_{0.333}$}. As $x$ further increases to 50.0%, the $T_C$ increases slightly. The variable-temperature SHG signals for the CP solid solutions were measured to check their NLO-switching function. As the ratio of **2** increases from 1.0 to 33.3%, all the CP solid solutions behave as a high-contrast thermoresponsive NLO switch accompanied by a continuously declining $T_S$ (Fig. 9), in accordance with the DSC measurements. Meanwhile, the SHG-active area expands across the minimum $T_C$ (Supplementary Fig. 18), leading to a curves-crossed phenomenon for the ratios of 33.3 and 50.0% (Fig. 7). Although the SHG intensity decreases with an increased ratio of **2**, the SHG signal remains strong enough in comparison with that of KDP. When the ratio of **2** rises to 0.60, no detectable SHG signals are observed before/after

phase transition. Overall, through the dual solid solution approach, the $T_S$ can be finely tuned below and above room temperature (roughly ranging from 273 to 328 K). The lowering of $T_S$ for these CP solid solutions can be attributed to the weakening interaction between the host {$[Cd(SCN)_3]^-$}$_\infty$ chains and the guest ammonium cations.

## Discussion

In summary, by combining the thermoresponsive NLO-switching function of phase-change material and the dual solid solution approach within the CP system, via mixed melting treatment, we provide a method for producing thermoresponsive NLO switch with highly tunable on-off temperature. In addition, an abnormal shortening of Cd–S bond in the melting state was revealed by in situ variable-temperature EXAFS analysis, owing to the tension contraction of the anionic chain. We are currently extending such a synthetic technique to other CP solid solutions with precisely tunable functions.

## Methods

**Materials**. All chemicals were commercially available and used without further purification. FT-IR spectra of KBr pellets were recorded on a Perkin-Elmer Spectrum One FT-IR spectrometer from 4000 to 400 cm⁻¹. PXRD patterns (Cu-Kα) were collected on a Bruker Advance D8 θ–2θ diffractometer. Thermogravimetric analyses (TGA) were carried out on a TA Q50 system at a heating rate of 10 K min⁻¹ in a N₂ flow. Differential scanning calorimeter (DSC) measurements were performed on a TA Q2000 instrument, at a heating/cooling rate of 10 K min⁻¹. Variable-temperature SHG experiments with a programmable temperature control system were executed by Kurtz-Perry powder SHG test using an Nd:YAG laser (1064 nm). The values of the NLO coefficients for SHG were determined by the comparison with commercialized KDP (KH₂PO₄).

**Syntheses of CPs 1 and 2**. A mixture of CdSO₄ (1 mmol), KSCN (3.0 mmol) and N,N-dimethylisopropylamine (1.0 mmol) was dissolved in aqueous H₂SO₄ (1.0 mmol/g) under stirring. The resultant clear solution was allowed to stand at room temperature. After several hours later, colorless column-shaped crystals of **1** were deposited from the solution, in ca. 81% yield based on Cd. When a similar procedure was performed but with N,N-diethylmethylamine in place of N,N-dimethylisopropylamine, column-shaped crystals of **2** were obtained in ca. 77% yield based on Cd. The PXRD patterns for grinded crystals indicated that their experimental pattern match well with the simulated ones (Supplementary Fig. 2). IR data (KBr, cm⁻¹) for **1**: 3443(m), 3110(m), 2987(m), 2974(m), 2131(s), 2107(s), 1633(m), 1462(s), 1437(m), 1409(m), 1385(m), 1348(m), 1318(m), 1148(m), 1081(m), 989(m), 920(m), 882(m), 755(m); 721(m), 616(m), 547(m). for **2**: 3450(m), 3121(m), 2998(m), 2981(m), 2091(s), 1638(m), 1476(m), 1449(s), 1389(m), 1364(m), 1146(s), 1041(m), 1001(m), 874(m), 829(m), 764(m), 664(m), 617(m).

**Syntheses of the solid solutions of CPs 1 and 2**. The mixture of **1** and **2** in a mole ratio of (1—x):x was grinded and mixed well into a small glass bottle, then placed in an oven at a temperature of 120 °C for 3 h, as a melting state. After slow cooling to room temperature, colorless CP solid solutions adhered to the bottle bottom were obtained.

**Single-crystal X-ray crystallography**. The single-crystal X-ray diffraction intensities for **1** and **2** at 296 K were collected on a Smart Apex II CCD diffractometer, while those for **1** at 348 K were collected on a Rigaku Oxford SuperNova diffractometer. Both diffractometers are equipped with a graphite-monochromated Mo-Kα radiation (λ = 0.71073 Å). Absorption corrections were applied by using multi-scan program SADABS or CrysAlisPro (Agilent Technologies) software[44,45]. The structures were solved with direct methods and refined with a full-matrix least-squares technique with the SHELXTL program package[46]. Anisotropic atomic displacement parameters were applied to all non-hydrogen atoms except the quite disordered (i-PrNHMe₂)⁺ cations of **1** at 348 K. All H atoms were positioned geometrically and included in the refinement in the riding-model approximation. Crystallographic data and structural refinements for **1** and **2** at different temperatures are summarized in Supplementary Table 1. The selected bond lengths of them are listed in Supplementary Tables 2 and 3, respectively.

**Extended X-ray absorption fine structure (XAFS) measurements**. All the acquired XAFS spectra of the samples and references were measured at the beamline BL 14W1 of the Shanghai Synchrotron Radiation Facility (SSRF). Samples that were pressed into thin sheets were positioned at 45° to the incident beam in the sample-holder. The typical energy of the storage ring and the electron current were 3.5 GeV and ~220 mA in the top-up mode, respectively. The monochromatized white light via a Si (311) double-crystal monochromator was calibrated with Cd(aq) as reference. All the spectra were measured with a Lytle detector in a fluorescence mode. A software package of Demeter[47] was employed to analyze the XAFS data. First of all, the spectra were normalized by Athena, and the subsequent shell fittings were performed with Artemis. All fittings were executed in R-space, and Fourier transformed (FT) using k³ weighting was adopted to the χ(k) function. According to the fitting results of the Cd(aq) reference, the amplitude reduction factor (S₀²) can be estimated as 0.839. By fitting the experimental peaks with theoretical amplitude, coordination parameters for the test samples were evaluated. Moreover, a wavelet transform technique was carried out through the Igor pro script developed by Funke et al.[48], so as to investigate the first-shell backscattering atoms and detect light as well as heavy scatters. Due to the fine resolution in both wavenumbers k and radial distribution function R, this qualitative analysis, as a complement to overcome the limitation of FT analysis, was mainly focused on the nature of the backscattering atoms and the bond lengths. For a better resolution in the wave vector k, the Morlet wavelet was set as basis mother wavelet and the parameters of η = 8 and σ = 1 were used.

**Classical MD simulation**. The Forcite program as part of the Materials Studio 5.5 package[49], was used to perform the classical MD simulations for **1**. Periodic models based on the experimental crystal data and the Universal forcefield were adopted for all the simulations. The constant-volume and temperature (NVT) ensemble with Nose thermostat and random initial velocities method were used to simulate the dynamic processes, with the unit-cell parameters of initial models being fixed while all atoms and coordinates being set in free motion. The charge equilibration (QEq) method[50] was employed to estimate the partial atomic charges of all atoms. The buffer widths of the electrostatic interactions and the van der Waals interactions evaluated by the Ewald summation method were both set as 0.5 Å. The total simulation time was 4 ns, with a time step of 1 fs.

## Data availability

The data that support the findings of this study are available from the corresponding author upon reasonable request.

## Code availability

The X-ray crystallographic coordinates for structures reported in this Article have been deposited at the Cambridge Crystallographic Data Centre (CCDC), under deposition numbers CCDC 1962900, 1962901 and 1962902. These data can be obtained free of charge from The Cambridge Crystallographic Data Centre via www.ccdc.cam.ac.uk/data_request/cif.

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

## Acknowledgements

This work was supported by the NSFC (21661004, 21671202 and 21821003), the Funding Project for Academic/Technical Leaders of Jiangxi Province (20172BCB22021), the NSF of Jiangxi Province (20192ACB20013), and the Local Innovative and Research Teams Project of Guangdong Pearl River Talents Program (2017BT01C161). C.-T.H. acknowledges the Young Elite Scientists Sponsorship Program by CAST. We thank the BL14W1 in SSRF for XAS measurement.

## Author contributions

Z.Y.D., S.Y.Z. and W.X.Z. conceived the idea for the project; S.Y.Z. and X.S. conducted material syntheses and routine characterizations such as TGA and IR spectra; Y.Z. and X.S. performed the data collection of single-crystal X-ray diffraction, PXRD and DSC measurements; Y.Z. and Q.Y.L. conducted the SHG measurements; C.T.H. and Z.Y.D. performed MD simulation, XAFS measurement and analyses; Z.Y.D., C.T.H., W.X.Z. and X.M.C. drafted the manuscript. S.Y.Z., X.S. and Y.Z. contributed equally to this work. All authors discussed and commented on the manuscript.

## Competing interests

The authors declare no competing interests.
