## [Peer Review File · Nature Communications]

Reviewers' comments:

Reviewer #1 (Remarks to the Author):

The switching of nonlinear optical signal by external stimuli are very interesting for their potential applications in sensing, signal processing and data storage. Until now, one most feasible method is via the thermal stimulation in a reversible manner. However, one main problem for practical application is how to regulate the switch temperature to the desired value, especially for the temperature range near the room temperature. In this manuscript, the authors put forward a novel dual solid solution strategy to solve this problem. They initially prepare a hexagonal perovskite CP as a high-contrast thermoresponsive NLO switch. Furthermore, by taking advantage of a synergistic dual solid solution effect, the melt mixing of it with its analogue afforded solid solutions with highly tunable switch temperature just above and below the room temperature. Overall, this paper is well written, and the switch and regulation mechanisms have been clearly clarified, supported using experimental variable-temperature testing technology of X-ray structure analysis, DSC, PXRD, SHG and XAFS measurements as well as theoretical molecular dynamics simulation. I believe their findings would be an interesting and valuable contribution to the field of NLO switches, and thus strongly recommend its publication in the journal of Nature Communication. Some minor comments and questions that the authors should reply before publication are listed below.

1. Line 55: The authors said that "To find a CP structure that can be melted before decomposition, we noticed that low-dimensional negatively charged coordination polymers templated with charge-balanced guest cations may be good candidates". Please provide some related literatures as reference.
2. Line 80, Chart 1: The word "isomeric" in the caption of Chart 1 may be deleted. I guess that non-isomeric ones in some case may also be suitable for the preparation of such solid solutions.
3. Lines 135 and 143: "Fig.s" should be revised as "Fig.".
4. Lines 198: The variable-temperature XANES analyses provide a lot of useful information about the melting state. But, what does "Cd(aq)" means? This point needs to be made clear.

Reviewer #2 (Remarks to the Author):

Comments on "Molecule-based nonlinear optical switch with highly tunable on-off temperature using a dual solid solution approach"

In this work, an alloy-like NLO switch with tunable TS via a dual solid solution approach within the coordination polymer (CP) system is reported. This NLO switch originates from a noncentrosymmetric to centrosymmetric phase transition at around 328 K. it can afford TS-tunable CP solid solutions in a range of about 273–328 K merely by varying the component ratio. This is very interesting work for NLO switching materials. It is publishable after authors answer the following questions:

- 1) It is mentioned that 1 and 2 are stable up to 416 K, which can be found from Fig.S1. But it is not so clearly seen that they are stable up to 416 K. Please give details of how to derive this 416K from Fig. S1?
- 2) How is the performance of SHG of 1 or 2 compared to the commercialized KTIPO_4 , LiB_3O_5 , LiNbO_3 , KNbO_3 , and $\text{Ba}_2\text{Na}(\text{NO}_3)_5$?
- 3) Please explain why the curves crossed for the ratio of 33.3% and 50% (Fig. 7).
- 4) What is the optimal component ratio for the best SHG intensity with high Ts?

Manuscript number: NCOMMS-19-37606

MS Type: Article

Title: "Molecule-based nonlinear optical switch with highly tunable on-off temperature using a dual solid solution approach"

Correspondence Author: Prof. Dr. Zi-Yi Du

Dear Reviewers,

Thank you very much for your comments and suggestions on our manuscript. We have revised the manuscript, taking all the comments and suggestions into consideration, and now submit a revised version. The comments and corrections have been highlighted in red. Our point-by-point responses to the referees are given below.

Sincerely,
Zi-Yi Du

Reviewers' comments:

Reviewer #1 (Remarks to the Author):

The switching of nonlinear optical signal by external stimuli are very interesting for their potential applications in sensing, signal processing and data storage. Until now, one most feasible method is via the thermal stimulation in a reversible manner. However, one main problem for practical application is how to regulate the switch temperature to the desired value, especially for the temperature range near the room temperature. In this manuscript, the authors put forward a novel dual solid solution strategy to solve this problem. They initially prepare a hexagonal perovskite CP as a high-contrast thermoresponsive NLO switch. Furthermore, by taking advantage of a synergistic dual solid solution effect, the melt mixing of it with its analogue afforded solid solutions with highly tunable switch temperature just above and below the room temperature. Overall, this paper is well written, and the switch and regulation mechanisms have been clearly clarified, supported using experimental variable-temperature testing technology of X-ray structure analysis, DSC, PXRD, SHG and XAFS measurements as well as theoretical molecular dynamics simulation. I believe their findings would be an interesting and valuable contribution to the field of NLO switches, and thus strongly recommend its publication in the journal of Nature Communication. Some minor comments and questions that the authors should reply before publication are listed below.

1. Line 55: The authors said that “To find a CP structure that can be melted before decomposition, we noticed that low-dimensional negatively charged coordination polymers templated with charge-balanced guest cations may be good candidates”. Please provide some related literatures as reference.

Re: Thanks for the suggestion. We have provided some related literatures as reference (See references 24-26): Also, the reference numbers in the text and the Reference Section have been updated.

2. Line 80, Chart 1: The word “isomeric” in the caption of Chart 1 may be deleted. I guess that non-isomeric ones in some case may also be suitable for the preparation of such solid solutions.

Re: Thanks for the suggestion. Indeed, non-isomeric ones in some case may also be suitable for the preparation of such solid solutions. We have deleted the word “isomeric” in the caption of Chart 1.

3. Lines 135 and 143: “Fig.s” should be revised as “Fig.”.

Re: Thank you for the kind reminding. We have revised it.

4. Lines 198: The variable-temperature XANES analyses provide a lot of useful information about the melting state. But, what does “Cd(aq)” means? This point needs to be made clear.

Re: Thank you for the suggestion. We have revised “Cd(aq)” as “[Cd(H₂O)₆]²⁺ in aqueous solution (Cd_{aq})”.

Reviewer #2 (Remarks to the Author):

Comments on “Molecule-based nonlinear optical switch with highly tunable on-off temperature using a dual solid solution approach”

In this work, an alloy-like NLO switch with tunable TS via a dual solid solution approach within the coordination polymer (CP) system is reported. This NLO switch originates from a noncentrosymmetric to centrosymmetric phase transition at around 328 K. it can afford TS-tunable CP solid solutions in a range of about 273–328 K merely by varying the component ratio. This is very interesting work for NLO switching materials. It is publishable after authors answer the following questions:

1) It is mentioned that 1 and 2 are stable up to 416 K, which can be found from Fig.S1. But it is not so clearly seen that they are stable up to 416 K. Please give details of how to derive this 416 K from Fig. S1?

Re: Thanks for the kind reminding. Because of our carelessness, the unit of measurement for temperature shown in Fig. S1 was the Celsius degree (°C), which should plus 273 to convert to the Kelvin temperature (K) we mentioned in the text. We have updated a revised Fig. S1 in the Supporting Information.

2) How is the performance of SHG of 1 or 2 compared to the commercialized KTiOPO₄, LiB₃O₅, LiNbO₃, KNbO₃, and Ba₂Na(NbO₃)₅?

Re: Thanks for the comment. The SHG of KH₂PO₄ (KDP), KTiOPO₄ (KTP) and LiB₃O₅ (LBO) can be employed under the irradiation of high-power laser, while that

of LiNbO_3 (LN), KNbO_3 (KN), and $\text{Ba}_2\text{Na}(\text{NbO}_3)_5$ (BNN) are often restricted to the irradiation of low-power laser for their relatively low laser damage threshold. For compound **1**, as an organic-inorganic hybrid compound, a relatively high laser damage threshold can be expected. For compound **2**, it shows no SHG effect because it crystallizes in the centrosymmetric space group in its both phases.

On the other hand, from the perspective of wavelength, most of the above-mentioned commercialized NLO materials as well as compound **1** are usually used for the visible generation, while the LiB_3O_5 is unique for the UV generation because the B-O bond is beneficial to the transmission of UV light.

The SHG measurement by the Kurtz and Perry method indicated that compound **1** is phase-matchable in the visible region, and the intensity of SHG signal for it is a little larger than that of KDP, but smaller than that of the very excellent KTiOPO_4 . The uniqueness of compound **1** is that it can function as a NLO switch originating from a noncentrosymmetric to centrosymmetric phase transition at around 328 K, and as further demonstrated in this work, such NLO switching behavior could be tuned by varying the component ratio of the solid solution of **1** and **2**.

3) Please explain why the curves crossed for the ratio of 33.3% and 50% (Fig. 7).

Re: Thanks for the suggestion. The NLO switch temperature of the CP solid solutions $\{\mathbf{1}_{1-x}\mathbf{2}_x\}$ depends on their solid-solid phase transition temperature. Hence, to understand why the curves crossed for the ratios of 33.3% and 50.0% shown in Fig. 7, the relationship of the solid-solid phase transition temperature (T_C) and the component ratio (x) for the CP solid solutions $\{\mathbf{1}_{1-x}\mathbf{2}_x\}$ is investigated in a full range, with x increasing from 0 to 100%. The following chart presents three possible model plots of T_C versus x for the general solid solutions $\{\mathbf{A}_{1-x}\mathbf{B}_x\}$.

For model I and II, the T_C dwells between $T_{C(A)}$ and $T_{C(B)}$, and it increases continuously upon increasing x . Obviously, there is no curves-crossed phenomenon. For model III, with the continuous increase of x , the T_C firstly decreases to a minimum, then increases to a maximum. The U- or V-shaped relationship makes it possible to form a curves-crossed phenomenon. As far as the experimental plot of T_C versus x for the CP solid solution $\{\mathbf{1}_{1-x}\mathbf{2}_x\}$ (see the updated Fig. S18), it corresponds to that of model III. Moreover, the SHG-active area expands across the minimum T_C (Fig. S18), leading to a curves-crossed phenomenon for the ratios of 33.3% and 50.0% shown in Fig. 7. We have added a related discussion on this phenomenon in the text, and updated the previous Fig. S18 as follows.

Fig. S18. Solid-solid phase transition temperatures (upon heating) as a function of x in the solid solutions $\{1_{1-x}2_x\}$. The SHG active area of the solid solutions is shaded in light blue.

4) What is the optimal component ratio for the best SHG intensity with high T_s ?

Re: The best SHG intensity with high T_s is that of the pure compound **1**. As shown in Fig. 7, it is clear that the SHG intensity decreases as the ratio of **1** decreases.

REVIEWERS' COMMENTS:

Reviewer #2 (Remarks to the Author):

1) It is mentioned that 1 and 2 are stable up to 416 K, which can be found from Fig.S1. But it is not so clearly seen that they are stable up to 416 K. Please give details of how to derive this 416 K from Fig. S1?

Re: Thanks for the kind reminding. Because of our carelessness, the unit of measurement for temperature shown in Fig. S1 was the Celsius degree (oC), which should plus 273 to convert to the Kelvin temperature (K) we mentioned in the text.

We have updated a revised Fig. S1 in the Supporting Information.

My question is not only for your mistake of the unit in the previous Fig. S1, but also wondering how you get the stable temperature of 416K. Was it extrapolated from the experimental measurements or calculated from any formula? I want to know how you get this value of 416K.

Manuscript number: NCOMMS-19-37606A

MS Type: Article

Title: "Molecule-based nonlinear optical switch with highly tunable on-off temperature using a dual solid solution approach"

Correspondence Author: Prof. Dr. Zi-Yi Du

Dear Reviewers and Editors,

Thank you very much for your comments. We have revised the manuscript, taking all the comments and suggestions into consideration, and now submit a revised version. The comments and corrections have been highlighted in red. Our point-by-point responses to the referees' and editors comments are given below.

Sincerely,
Zi-Yi Du

REVIEWERS' COMMENTS:

Reviewer #2 (Remarks to the Author):

1) It is mentioned that 1 and 2 are stable up to 416 K, which can be found from Fig.S1. But it is not so clearly seen that they are stable up to 416 K. Please give details of how to derive this 416 K from Fig. S1?

Re: Thanks for the kind reminding. Because of our carelessness, the unit of measurement for temperature shown in Fig. S1 was the Celsius degree (oC), which should plus 273 to convert to the Kelvin temperature (K) we mentioned in the text.

We have updated a revised Fig. S1 in the Supporting Information.

My question is not only for your mistake of the unit in the previous Fig. S1, but also wondering how you get the stable temperature of 416K. Was it extrapolated from the experimental measurements or calculated from any formula? I want to know how you get this value of 416K.

Re: Thanks for your comment. Here, the stable temperature of 416 K is extrapolated from the experimental measurement. TGA is a rough but commonly used method for the thermal stability evaluation. We usually judge the initial (on-set) decomposition temperature, at which the weightlessness begins, as the stable temperature. Since the initial decomposition temperature of TG is often influenced by many factors such as the heating rate, the gas flow rate and the sample size, the value of it is only a rough assessment for the stable temperature. Taking this into consideration, we revised the description "stable up to 416 K" as "stable up to about 416 K".

Also, we have reviewed the changes in the attached copy of our manuscript raised by the Editors, and tried our best to edit our manuscript to comply with the format requirements of *Nature Communication* and to maximize the accessibility. **For all changes, we have used the 'track changes' feature in the revised Microsoft Word file.** The major changes are summarized as follows.

1. The abstract section has been revised to be less than (or equal to) 150 words. In addition, we have removed abbreviations from the abstract and instead wrote them out in full wherever they appear in the abstract.
2. We have avoided using terms like 'for the first time', 'new' or 'novel' etc. throughout the manuscript.
3. We have labeled 'Introduction' and 'Discussion' as headings, and all subheadings in the 'Results' and 'Methods' sections have been revised to be <60 characters (including spaces) to comply with the article templates of *Nature Communication*. The subheadings of 'Results' have been placed on a separate line.
4. We have labeled the previous "Scheme 1" and "Chart 1" as "Fig. 1" and "Fig. 2", respectively. Legends of all figures have been revised to begin with a brief title describing the whole figure without punctuation.
5. We have formatted 'W/g' as 'W g⁻¹' in the new Fig. 3 and Supplementary Fig. 17.
6. We have revised the Reference section to comply with the Nature referencing style.
7. A structural figure with probability ellipsoids for compound **2** at 348 K has been added to the updated Supplementary Fig. 6 (see the Supplementary Information).

In addition, for the two-sentence editor's summary, I fully approve the draft summary provided by the editor.